# Synergy of Muscle and Cortical Activation through Vojta Reflex Locomotion Therapy in Young Healthy Adults: A Pilot Randomized Controlled Trial

**DOI:** 10.3390/biomedicines11123203

**Published:** 2023-12-01

**Authors:** Juan Luis Sánchez-González, Emiliano Díez-Villoria, Fátima Pérez-Robledo, Ismael Sanz-Esteban, Inés Llamas-Ramos, Rocío Llamas-Ramos, Antonio de la Fuente, Beatriz María Bermejo-Gil, Ricardo Canal-Bedia, Ana María Martín-Nogueras

**Affiliations:** 1Department of Nursing and Physiotherapy, Instituto de Investigación Biomédica de Salamanca (IBSAL), University of Salamanca, 37008 Salamanca, Spain; juanluissanchez@usal.es (J.L.S.-G.); inesllamas@usal.es (I.L.-R.); rociollamas@usal.es (R.L.-R.); beatriz.bermejo@usal.es (B.M.B.-G.); anamar@usal.es (A.M.M.-N.); 2Centro de Atención Integral al Autismo-InFoAutismo, INICO-Instituto Universitario de Integración en la Comunidad and Investigación Biomédica de Salamanca (IBSAL), University of Salamanca, 37008 Salamanca, Spain; emid@usal.es (E.D.-V.); rcanal@usal.es (R.C.-B.); 3Physical Therapy and Health Research Group, Department of Physiotherapy, Faculty of Sport Sciences, Universidad Europea de Madrid, Villaviciosa de Odón, 28670 Madrid, Spain; ismael.sanz@universidadeuropea.es; 4University Hospital of Salamanca, 37007 Salamanca, Spain; 5Department of Physiology and Pharmacology, Institute of Neurosciences of Castilla and León (INCyL), University of Salamanca, Avenida Alfonso X El Sabio s/n, 37007 Salamanca, Spain; jfuente@usal.es; 6Department of Nursing and Physiotherapy, Institute of Neurosciences of Castilla and León (INCyL), University of Salamanca, 37008 Salamanca, Spain

**Keywords:** Vojta Therapy, neurorehabilitation, muscle activity, cortical activity

## Abstract

Background: Vojta Therapy is a neurorehabilitation therapy that allows to activate reflex movement patterns. The scientific literature has shown its ability to generate muscle contractions. The activation of brain neural networks has also been proven. However, the relationship between these processes has not yet been demonstrated. For this reason, the aim of this study is to verify brain activation produced by recording with near-infrared spectroscopy and its relationship with muscle activation produced in the abdominal muscles recorded with surface electromyography. Methods: A total sample of 27 healthy subjects over 18 years of age was recruited. An experimental study on a cohort was conducted. Two experimental conditions were considered: stimuli according to the Vojta protocol, and a control non-stimuli condition. Abdominal muscle activation was measured using surface electromyography, and the activation of the motor cortex was assessed with near-infrared spectroscopy. Results: In relation to the oxygenated hemoglobin concentration (HbO), an interaction between the stimulation phase and group was observed. Specifically, the Vojta stimulation group exhibited an increase in concentration from the baseline phase to the first resting period in the right hemisphere, contralateral to the stimulation area. This rise coincided with an enhanced wavelet coherence between the HbO concentration and the electromyography (EMG) signal within a gamma frequency band (very low frequency) during the first resting period. Conclusions: The results underscore the neurophysiological effects on the brain following tactile stimulation via Vojta Therapy, highlighting increased activity in pivotal areas essential for sensory processing, motor planning, and control. This activation, particularly evident in the Vojta stimulation group, aligns with previous findings, suggesting that tactile stimuli can not only evoke the intention to move but can also initiate actual muscle contractions, emphasizing the therapy’s potential in enhancing innate locomotion and rolling movements in patients with neurological disorders.

## 1. Introduction

Vojta Therapy (VT) was developed by Czech professor and child neurologist Vaclav Vojta. In 1960, Vojta observed that sustained stimulation of peripheral pressure elicited a stereotyped generalized motor response, manifested as a pattern of tonic muscle contractions on both sides of the neck, trunk, and limbs, resulting from spatial summation that leads to improved postural control [1]. This methodology targets ontogenetic postural function and automatic postural control [2], upon which various environmental aspects will later act. The so-called “reflex locomotion” does not refer to neuronal regulation but is associated with therapeutically applied external stimuli and their automatic movement responses.

These proprioceptive stimuli generate reflex muscle contraction according to a determined and blocked ontogenetic pattern in children with neurological disorders. For this reason, its therapeutic use has been expanding over recent years.

The Vojta Method or Vojta Reflex Locomotion Therapy (VRLT) is a rehabilitation method for neuromusculoskeletal pathologies widely used in Europe. Its development is based on the motor ontogenesis concept and tries to trigger innate motor reactions (reflex locomotion patterns) in the trunk and limbs from defined tactile and proprioceptive stimuli, starting from specific postures [3,4]. It is an active therapy, in which the central axes of the treatment are the therapist and the patient; thus the patient must be concentrated. The main indication for this therapy is focused on children with motor disorders and infants at risk of cerebral palsy [5,6,7]. It has also been used in adults with neurological and motor impairment and in adults who have suffered a stroke [8]. Other indications are neurological pathologies such as multiple sclerosis [9,10,11] or even peripheral lesions, spina bifida, congenital malformations or orthopedic problems [12,13].

Many authors have tried to answer the action mechanism of Vojta Therapy. Specifically, using surface polyelectromyography in healthy subjects, an activation of the musculature transmitted through the proprio-spinal tract fibers has been described [14], as well as changes in subcortical regions (putamen) and cerebellum, structures that play an important role in motor control [15], and the activation of cortical areas such as supplementary motor areas (SMA) and premotor areas (PMA) (Brodmann areas BA6 and BA8), superior parietal cortex (BA5, BA7) and posterior cingulate cortex (BA23, BA31) [16] after this therapy application.

The objective measurement of the Vojta Therapy effects has been extensively investigated. One of the procedures to demonstrate this evidence is the use of surface electromyography (sEMG) to observe muscle contractions [17,18,19,20]. Several studies implemented in children and adults with neurological disorders have used this technology to test the efficacy of Vojta Therapy. Studies have also been implemented in healthy subjects such as those of Gajewska et al. [14], who have recorded bilateral deltoid and rectus femoris contraction using a sEMG; Sanz-Esteban et al. [21] who observed effects in the finger extensor muscle during manual Vojta Therapy stimulation of the reflex rolling; and Pérez-Robledo [19] who observed effects in the rectus abdominis, external oblique, internal oblique, and serratus anterior muscles. In addition, Perales-López and Fernández-Aceñero [22] recorded the muscle contraction of the common extensor digitorum to evidence the transferable potential of Vojta Therapy using teleneurorehabilitation. This sEMG technique has also been used in orthopedic pathology and multiple sclerosis [17,18,20].

Another technology widely used nowadays is near-infrared spectroscopy (NIRS), a non-invasive technique that offers the advantage of monitoring cerebral oxygenation, giving us a picture of the metabolic and hemodynamic status of multiple regions [23,24,25]. Functional near-infrared spectroscopy (fNIRS) operates within a wavelength range of 700 nm to 1000 nm. Within this range, the absorption of chromophores such as oxygenated and deoxygenated hemoglobin is maximized, while the absorption of other compounds, like water molecules, is minimized [25]. Several types of fNIRS are currently in use: time-domain devices, frequency-domain devices, and continuous wave devices. Most of the systems available on the market are continuous waves. These devices are based on the principle of infrared light emission at two specific wavelengths, its absorption by chromophores (oxyhemoglobin and deoxyhemoglobin), and its reflection. fNIRS light can penetrate bone structures and several millimeters into brain tissue where, according to the Beer–Lambert law, light absorption is directly proportional to the concentration of chromophores. The attenuation of reflected light represents information about regional brain oxygen saturation (rSO2) and the balance between oxygen supply and oxygen consumption, which makes fNIRS a very sensitive technology to changes in brain oxygenation [25]. Based on the above, fNIRS could address ideal neuromonitoring requirements, detect brain tissue at risk for secondary injury, and complement or even replace current invasive practices [26,27].

It has already been shown that fNIRS has an essential role in the diagnosis and treatment of various neurological disorders, such as Alzheimer’s dementia, epilepsy, neurotrauma, and stroke rehabilitation [28,29,30], being able to detect changes in the premotor cortex of the affected hemisphere associated with motor recovery after rehabilitation [28]. However, despite all this evidence through all these measurement techniques, to date, there is no study that has used surface electromyography and near-infrared spectroscopy to measure the coherence between the two signals.

For all these reasons, the present study is proposed with the aim of testing muscle activation and brain oxygenation resulting from Vojta Therapy in healthy subjects.

## 2. Materials and Methods

### 2.1. Design

A randomized controlled trial was conducted. The project was implemented at the Functional neuroimaging laboratory for the analysis of live social interaction at the Comprehensive Autism Care Center (infoAutismo) of the University Institute for Community Integration (Salamanca, Spain). The clinical trial received approval from the Ethics Committee of University of Salamanca (record number 757/2022) and was carried out in accordance with the Declaration of Helsinki.

The CONSORT 2010 Statement (Consolidated Standards of Reporting Trials) was followed [31], and the clinical trial was registered in ClinicalTrials.gov with the registration number NCT05170906.

### 2.2. Participants Recruitment and Eligibility Criteria

Healthy subjects between 18 and 30 years old were recruited from the University of Salamanca. Recruitment was performed via email and through the usual channels of information in a university community between November 2022 and January 2023. The inclusion criteria were: (a) being male; (b) being over 18 years of age; (c) not being familiar with Vojta Therapy. The exclusion criteria were as follows: (a) suffering from neuromuscular pathology; (b) subjects with previous abdominal or head surgeries; (c) sensory alterations or the presence of inflammatory diseases; (d) subjects with pharmacological treatment that could affect the nervous system were also excluded.

### 2.3. Allocation and Randomization

Participants were randomly assigned into two groups: a non-specific tactile input group (non-STI group) (*n* = 15) and a Vojta specific tactile input group (V-STI group) (*n* = 15) (Figure 1).

Concealed allocation was conducted with a computer-generated randomized table of numbers in a blocked proportion of 1:1 (GraphPad Software Inc., San Diego, CA, USA). A researcher, not involved in any other aspect of the experiment, carried out the randomization procedure. Individual and sequentially numbered index cards with the random assignment were folded in sealed opaque envelopes. The researcher opened the appropriate envelope after baseline data collection.

### 2.4. Blinding

None of the participants previously knew the groups or the area of the stimulus where the intervention was going to be applied (blinded participants). The physical therapist who performed the intervention was not involved in any other aspect of the research such as analysis or data collection and was the only person who knew the nature of the intervention. All assessments were recorded with a blinded assessor.

### 2.5. Procedure

All subjects were stimulated by an expert Vojta therapist. A standardized recording protocol was established for abdominal muscle and for motor cortex oxygenation recordings. The protocol was identical for both groups:10 min of absolute rest without recording.1 min of rest prior to stimulation with EMG and fNIRS recording (R1).4 min of right-sided styling stimulation with EMG and fNIRS recording (S1).1 min of rest with EMG and fNIRS (R2).4 min of stimulation of the left side with EMG recording and fNIRS (S2).1 min of rest with EMG recording (R3).10 min absolute rest without EMG or fNIRS recording.

### 2.6. Tactile Stimuli Location and Sham Stimuli Location

All the participants were placed face up on a stretcher to carry out the intervention. They took a comfortable position, which they could maintain during the treatment, with the upper and lower extremities extended along the body. The stimulus provided in the V-STI group was performed on the right side of the body, in the intercostal space located between the 7th and 8th rib, following the mamillary line. The direction of the stimulus was dorsal, medial, and cranial, in the direction towards the opposite shoulder. An approximate pressure of 1.4 and 1.8 kg/cm^2^ ± 200 g was applied. The physical therapist who performed the intervention trained this pressure prior to each intervention with a pressure algometer.

In the non-STI group, the sham stimulus was performed in an area of low receptor density located on the ventral side of the thigh, between the proximal and medial thirds. The pressure exerted was the same as in the V-STI group and followed a dorsal direction.

The stimulation applied in both groups consisted of sustained manual pressure over a specific area of the body. Throughout the therapy application, there were no movement artifacts that could have influenced the fNIRS and EMG recordings.

### 2.7. Electromyographic Recording (rEMG) and Cerebral Oxygenation Recording (rCO)

A Brainquiry electromyographic signal recording device, model PET4, was used. The device contains a programmable amplifier and an analog-to-digital converter and was configured as a double channel AC differential signal amplifier which minimizes electrical noise derived from movements and other artifacts. ADC was performed at 1 KHz sample rate. Ag/AgCl electrodes of 2 centimeters in diameter placed in pairs on each muscle at an inter-electrode distance of 3 cm and following the preferential arrangement of the muscle fibers were used to obtain the differential recording between the two. The electrodes were placed according to the Surface Electromyography for the Non-Invasive Assessment Muscles (SENIAM) criteria [32]. The muscles recorded were the external oblique and internal oblique muscles on both sides. Acquired signals were sampled at 1 KHz and filtered with a 60 Hz high-pass digital filter to reject signals of non-muscle origin.

Brain oxygenation was recorded using a NIRScout system (NIRx Medical Technologies LLC, Berlin, Germany); it consists of 16 light sources and 16 light detectors with wavelengths of 760 nm and 850 nm. The motor cortexes of both hemispheres were covered with a 128-position NIRScap with adapters and clamping washers using nine sources and nine detectors, thus forming a total of 18 channels. The distance between optodes was kept at 3 cm. Acquisition was performed in a relatively dark room in a relaxed environment. The device was calibrated for each participant prior to acquisition. Optical density data were recorded and transformed to changes in oxyhemoglobin and deoxyhemoglobin concentration in the different areas of interest (different areas of the motor cortex). The change in hemoglobin levels in millimoles (mM) indicates the change in chromophore concentration after stimulation relative to a resting phase. The modified Beer–Lambert Law was applied to convert optical density data into relative concentration changes in HbO and HHb.

### 2.8. rEMG and rCO Data Analysis

Data from EMG signal obtained as described before were sent to a computer equipped with Bioexplorer/Bioreview software PET4 (www.cyberevolution.com, accesed on 1 July 2023), which includes a variety of modules to process the signals and a file transfer module in standard CSV (comma separated values) format. A further cleaning step was performed manually over the data to remove some remaining artifacts, and finally, RMS values at 1 Hz epochs/bins were obtained from the data and exported to analysis software.

### 2.9. Statistical Analysis

The fNIRS data were processed for analysis using the Python library MNE and several custom python and R scripts. The change in oxyhemoglobin (HbO) signal was used as the primary measure of brain activity.

First, raw optical intensity measures (the amount of light detected by the detector optodes) were converted to optical density (the amount of light absorbed by tissue). (Second, to detect potential sources of noise or artifact, scalp coupling index (SCI) was calculated for all channels with a high-pass filter cutoff frequency of 1.2 Hz and a transition band of 0.1). The dataset was then resampled to 1 Hz, the same frequency of the EMG recordings, and a temporal derivative distribution repair was applied to remove baseline shift and spike artifacts. Then, data were converted to hemoglobin concentration based on the Beer–Lambert law using a standard differential pathlength factor of 6, and the band-pass was filtered (between 0.02 Hz and 0.3 Hz) to retain frequencies typically associated with the hemodynamic response to neural activity. Finally, a signal processing function to enhance negative correlations was applied, and resultant datasets were merged with EMG signals, normalized (converted z-scores by channel), and exported for further analysis using ANOVAs (or robust ANOVAs when significant deviation from the assumptions was verified).

Since the stimulation protocol for this study involves only one trial per participant and the use of two physiological signals, we have chosen to apply a wavelet-based analysis, a signal analysis technique that has been widely used in fNIRS signal processing due to its advantages compared to other analysis methods [33]. For example, the wavelet transform allows the analysis of signals at different time and frequency scales, can effectively help to reduce noise, can detect signals originating from different physiological causes (e.g., respiratory, cardiac, Mayer waves, etc.), and produces an intuitive graphical representation of the fNIRS signal on the time and frequency scale. Also, wavelet coherence analysis allows to assess the relationship between two time series in both the time and frequency domains, using measures like coherency.

A coherency index between mean HbO in left and right hemispheres and the external and internal oblique EMG was obtained for different frequency bands: VLFO (0.01 to 0.02 Hz; 0.02 to 0.04 Hz), LFO (0.04–0.15 Hz), respiration (0.16–0.6 Hz), and cardiac (0.8–1.5 Hz). Those scores were entered in several Yuen’s bootstrapped *t* tests to explore possible differences in coherence between groups.

All analysis were conducted in R [34] with MNE python [35] and rstatix, WaveletComp, biwavelet, lme4, and lsmeans packages.

## 3. Results

### 3.1. Sociodemographic Data

The sample consisted of a total of 30 patients of the 32 selected at the study onset (Figure 1). Three subjects were excluded due to the impossibility of performing cortical recording with fNIRS or problems with the EMG recording. The mean age of the whole sample was 23.63 ± 1.49. The mean age for the non-STI group was 23.86 ± 1.9. The mean age for the V-STI group was 23.4 ± 0.73. Sample features are summarized in Table 1.

### 3.2. Intervention Effects on Cortical Activity

The HbO concentration z-scores were entered in a repeated measures ANOVA with group as a between-subjects variable (V-STI and non-STI), and stimulation phase (R1, S1, R2, S2, R3) and hemisphere (left, right) as a within-subjects variables. No significant main effect of stimulation phase, hemisphere [*F* < 1], or group [*F*(1, 25) = 1.23; *p* = 0.28] were found. With respect to the interactions, only the stimulation phase x group was significant [*F*(4, 100) = 3.03; *p* = 0.02; ηp2 = 0.1], with *F* values less than one for the rest of the interactions. An exploration of the differences between HbO concentration in the stimulation phases for each group using robust paired *t*-tests revealed a significant increase for the V-STI group between the R1 and S1 phases in the right hemisphere [*t*(9) = −2.37; *p* = 0.04; *d* = 0.68] and between the R1 and R2 phases in the right hemisphere [*t*(9) = −2.54; *p* = 0.03; *d* = 0.49]. No other comparison was found to be significant. Robust paired *t*-tests (Yuen’s bootstrapped) showed no significant increased HbO concentration in V-STI with respect to non-STI in any phase (only the left R2 difference was close to significance, *p* = 0.13) (Figure 2 and Table 2).

### 3.3. Intervention Effects on Muscle Activity

For each EMG measure, we conducted a robust repeated measures analysis of variance (ANOVA) with group as a between-subjects variable (V-STI and non-STI) and stimulation phase as a within-subjects variable (R1, S1, R2, S2, R3).

In the case of the left external oblique (m1R), the analysis revealed no significant effect for phase [*F*(4, 10.64) = 1.4; *p* = 0.30] but a significant main effect for group [*F*(1, 12.93) = 12.01; *p* = 0.004] and group x phase interaction [*F*(4, 10.64) = 5.24; *p* = 0.014]. Robust paired *t*-tests (Yuen’s bootstrapped) showed significant increased electric muscle activity in V-STI with respect to non-STI in phase S1 [*t*(11.5) = −3.36; *p* = 0.01; *ξ* = 0.66; 95% CI: 0.18–0.97], R2 [*t*(12.9) = −3.51; *p* < 0.0001; *ξ* = 0.76; 95% CI: 0.25–0.97], S2 [*t*(14.13) = −2.86; *p* < 0.01; *ξ* = 0.64; 95% CI: 0.11–0.97], and R3 [*t*(16.98) = −2.04; *p* < 0.05; *ξ* = 0.47; 95% CI: 0.02–0.87], but not in baseline phase R1 [*t*(16.6) = −0.81; *p* = 0.35].

For the right external oblique (m2R), analysis revealed only a significant main effect for group [*F*(1, 12.25) = 5.59; *p* = 0.04], but no significant effect for phase [*F*(4, 10.24) = 0.8; *p* = 0.56] or group x phase interaction [*F*(4, 10.24) = 2.32; *p* = 0.13]. Robust paired *t*-tests (Yuen’s bootstrapped) showed significantly increased electric muscle activity in V-STI with respect to non-STI in phase R2 [*t*(14.4) = −2.58; *p* < 0.01; *ξ* = 0.72; 95% CI: 0.11–0.97] but not in baseline phase R1 [*t*(16.8) = −0.27; *p* = 0.77], S1 [*t*(10.4) = −2.08; *p* = 0.06], S2 [*t*(11.31) = −1.82; *p* = 0.06], or R3 [*t*(15.56) = −1.31; *p* = 0.14] (Table 3).

Although significant effects of stimulation on EMG are not confirmed, a differential pattern in the values between the two conditions is evident (see Figure 3). In the case of m1R, an increase in values during stimulation moments (S1 and S2) is observed compared to its previous baseline periods. In the case of m2R, this increase is more pronounced during the S2 stimulation phase.

### 3.4. Intervention Effects on Coherence of Cortical Activity and Muscle Activity

Figure 4 shows the global wavelet coherence between mean HbO concentration in the left and right hemispheres and the external (m1R) and internal (m2R) oblique EMG signal for the V-STI group and non-STI group.

Visual inspection of the coherence plots shows increased rates of coherence between the left hemisphere HbO signal and the EMG signals for the V-STI group in the VLFO frequency band (scale > 25), especially after the onset of the S1 period (right-sided stimulation). Such coherence is not appreciable for the non-STI group. The increased coherence for V-STI was statistically verified for left HbO—m2R—and right HbO—m1R—in the R2 period (see Table 4).

## 4. Discussion

In recent years, there has been significant growth in the number of research studies conducted to elucidate the effects of physiotherapy and the application of stimuli that can aid in improving patients with neurological disorders. This present study aims to demonstrate the neurophysiological effects on the brain following tactile stimulation administered with Vojta Therapy. While numerous therapeutic tools can induce effects on cerebral activation [36,37,38], there are few that demonstrate activation of crucial areas in the neurorehabilitation of neurological patients after the application of tactile sensory stimuli.

The presence of increased activity in key areas crucial for processes related to sensory processing, motor planning, and motor control is fundamental to cerebral neuroplastic processes [39,40].

fNIRS has become the imaging instrument with the best capabilities for observing the activity of the nervous system during the performance of actions, cognitive processes, or imagination. In the present study, fNIRS was used, which, like fMR, allows us to record cortical activity during a specific action. Both tools are based on monitoring the concentration of oxyhemoglobin and deoxyhemoglobin in the areas of interest in the brain. While it is true that fMRI has higher resolution, the use of fNIRS is highly reliable.

In a study by Sanz Esteban et al. [15], functional magnetic resonance imaging (fMRI) was used to depict brain activity during the application of a tactile and proprioceptive sensory stimulus to the pectoral area (between the seventh and eighth ribs). Significant activity was recorded in premotor areas, and these findings are consistent with the motor activity recorded in the current study in motor areas. In both cases, it is the subjects undergoing Vojta Therapy who exhibit motor activity during the application of the stimulus.

The increase in blood volume during sustained pressure from the stimulus applied with Vojta Therapy leads to changes in somatosensory and pontine areas on the contralateral side of the stimulus location [41]. There is a relationship between increased blood flow and motor activity in the subjects of the experimental group (V-STI group). In the current study, there is a clear trend concerning the increase in HbO concentration during the stimulation periods and at the beginning of the resting phase (S1, R1), most notably in the left hemisphere during stimulation of the right pectoral area. The stimulation shows a significant correlation between these moments and the contractile activity of the abdominal muscles, specifically the ipsilateral external oblique muscle on the side of stimulation.

The results obtained are consistent with those obtained by Sanz et al., 2021 [16], regarding the type of contraction and responses in brain activation. Martinek et al. reported significant activity in areas BA3, 4, and 6 of the cortexes following sensory stimulation with Vojta Therapy [42].

Studies conducted in recent years that recorded brain activation using EEG during the application of a tactile stimulus consistent with Vojta Therapy show a significant increase in activation in association with areas BA5 and 7, which are responsible for receiving tactile sensory stimulation. What is truly interesting is that after receiving this sensory stimulus, the intention to move is transmitted to BA 6, and in some subjects, motor action in BA 4 occurs innately [16,21]. In the present study, we have recorded similar brain activity data, showing an increase in neural synchronization in the VLOF frequency bands.

EEG activity is divided into different frequency bands based on the functional activity of the subject. Previous studies have shown statistically significant data in the theta and low alpha frequency bands. In the recording of brain activity with fNIRS, it is in the very-low-frequency band where significant results exist when comparing activity between resting periods one and resting two. Sanz et al. obtained statistically significant and relevant data when comparing brain activity during a pre-Vojta Therapy stimulation rest period with the same rest period (resting) after stimulation. The results of both studies coincide in the activity of the low-frequency waves activity during the resting period after Vojta stimulation [16]. This activity translates into an increase in coherence between brain activity and the muscle activity recorded in this study.

Over the past decade, several authors have studied the presence of slow oscillation waves and activity in cortical networks [43,44]. Studies by Hok [41] and Sanz in 2018 and 2021 [15,16] present data on brain activity in areas related to motor networks, specifically the reticular formation, supplementary motor area, cerebellum, basal ganglia, and thalamus. The thalamus is a structure associated with the reception of stimuli. Garrett T. Neske discusses the relationship between the cortico-thalamic network system and slow oscillation waves, which is a characteristic of this system. In this context, these very low-frequency waves play a role in controlling corticomotor networks [43]. All of this leads us to hypothesize that the subject’s position, stimulation, and reception of the stimulus induced by Vojta Therapy activate a system of cortico-thalamic neural networks, corticomotor systems, and various areas related to movement preparation and regulation, thereby generating a triple interaction between sensorimotor areas, cortical systems, and the motor response triggered by tactile stimulation.

Sensory stimuli can trigger the desire for movement and even lead the subject to perform actual muscle contractions that initiate movement synergies [45]. Some previous studies have linked brain activation during tactile stimulation to the recording of electromyographic activity in specific muscle groups [14]. Additionally, changes in muscle activity during inspiration in the diaphragm have been recorded in subjects stimulated in the pectoral area with Vojta Therapy [46].

During the application of stimulation in the pectoral area, there has been a significant recording of activity in the oblique abdominal muscles. These findings are consistent with previous studies by Gaweska, Sanz, and Pérez [14,16,21], which demonstrate muscle activity in areas involved in innate locomotion and rolling movements.

### Limitations

The main limitation of this study is determined by the fact that only a single measurement has been conducted using near-infrared spectroscopy. The main studies that use this tool to monitor cerebral blood flow employ multiple trials. The primary reason for conducting only one trial is so that the subjects who participated in the research would not learn about the technique and remain completely unaware of the therapy and group allocation. However, for future trials, this limitation will be considered, and more trials per subject will be conducted. Nevertheless, the results obtained show a clear indication of this synergy of activation between the motor cortex and muscular activity, so further investigation in this field of study will be necessary. Furthermore, it is necessary to note that the current intervention was conducted on healthy subjects and only male subjects. It has allowed us to know the correct way to perform these procedures, but this is essential to point this out because the comparison and potential benefits of such therapy on subjects with neurological damage could be influenced by numerous additional factors.

## 5. Conclusions

The application of Vojta Therapy elicits brain activity in essential areas for sensory stimulus processing, movement planning, and motor control. During the application of proprioceptive stimulation, a direct relationship occurs between brain activity and contractile muscle activity, revealing the interaction of sensorimotor cortical areas, motor areas, and subcortical structures that regulate movement. Stimulation carried out in accordance with Vojta Therapy induces low-frequency wave activity in premotor areas, suggesting that not only is the intention to move activated, but also real muscle contraction occurs. This relationship implies that the effects of Vojta Therapy in the treatment of patients with neurological disorders have significant potential to improve innate locomotion and rolling movement patterns.

## Figures and Tables

**Figure 1 biomedicines-11-03203-f001:**
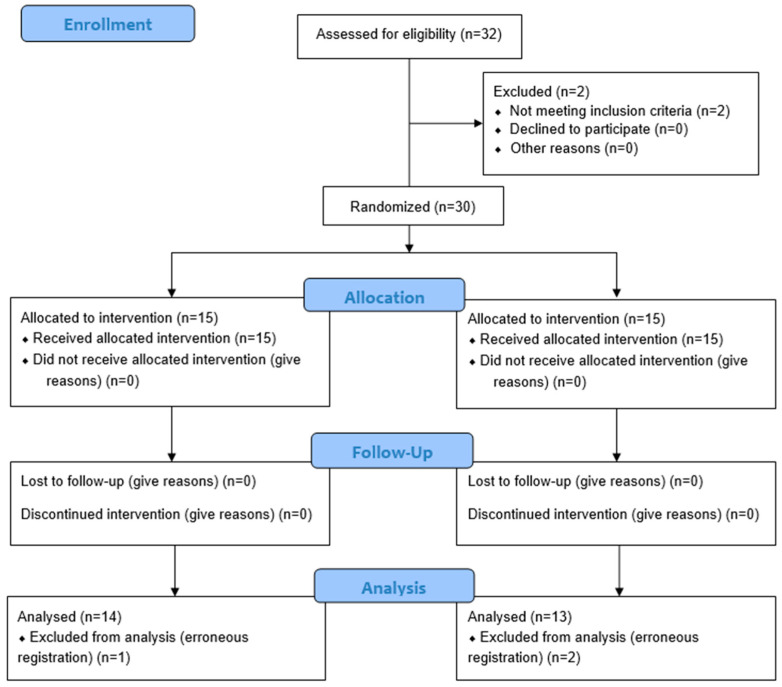
Flow chart.

**Figure 2 biomedicines-11-03203-f002:**
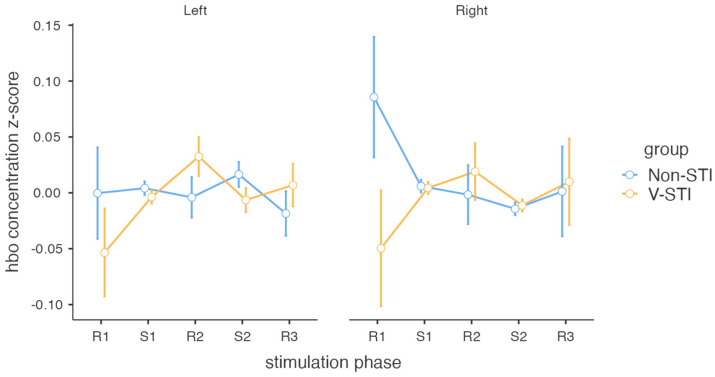
HbO concentration (z-score) by hemisphere and stimulation phase (**Left** and **Right**).

**Figure 3 biomedicines-11-03203-f003:**
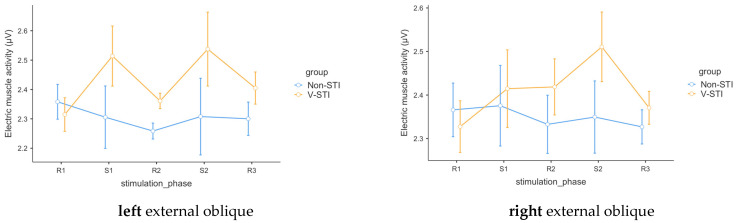
EMG by group. m1R = **left** external oblique; m2R = **right** external oblique.

**Figure 4 biomedicines-11-03203-f004:**
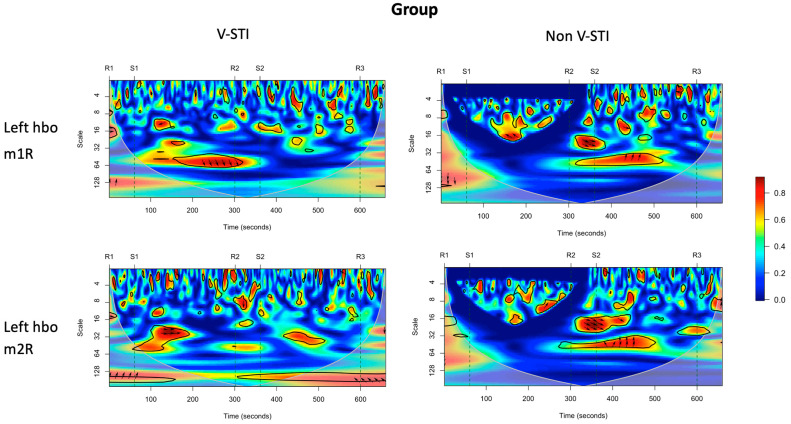
Wavelet coherence between mean HbO concentration in left (**top** panel) and right hemispheres (**bottom** panel) and left external (m1R) and right external (m2R) oblique EMG signal for V-STI group (**left**) versus non-STI group (**right**). Red areas represent stronger correlations between the two signals, while blue areas represent weaker correlations. The relative phase relationship is exhibited as arrows (with in-phase pointing right, anti-phase pointing left, and HbO (t) leads EMG(t) pointing straight down).

**Table 1 biomedicines-11-03203-t001:** Simple features.

Groups (*n*)	Age (years) Mean (±Standard Deviation)	Weight (Kilograms) Mean (±Standard Deviation)	Height (Centimeters) Mean (±Standard Deviation)
All sample	23.62 ± 1.54	72.77 ± 7.84	178.74 ± 3.93
Non-STI group (13)	24 ± 2.12	72.30 ± 9.49	179.76 ± 2.68
V-STI group (14)	23.28 ± 0.61	73.21 ± 6.27	177.78 ± 4.72

**Table 2 biomedicines-11-03203-t002:** Mean standardized HbO concentration by group, hemisphere, and stimulation phase.

Hemisphere	Group	Stimulation Phase	M	SE
Left	Non-STI	R1	−3.19 × 10^−4^	0.04088
		S1	0.00414	0.00611
		R2	−0.00405	0.01811
		S2	0.01642	0.01116
		R3	−0.01852	0.01982
	V-STI	R1	−0.05346	0.03939
		S1	−0.00373	0.00589
		R2	0.03239	0.01745
		S2	−0.00642	0.01075
		R3	0.00686	0.01910
Right	Non-STI	R1	0.08562	0.05390
		S1	0.00593	0.00566
		R2	−0.00168	0.02638
		S2	−0.01434	0.00560
		R3	0.00123	0.04013
	V-STI	R1	−0.04970	0.05194
		S1	0.00438	0.00546
		R2	0.01894	0.02542
		S2	−0.01128	0.00539
		R3	0.00991	0.03867

Note. M = mean; SE = Standard deviation.

**Table 3 biomedicines-11-03203-t003:** Mean EMG muscle activity by group and stimulation phase.

Group	Stim Phase	Left External Oblique		Right External Oblique	
		M	SE	M	SE
Non-STI	R1	2.36	0.0593	2.37	0.0615
	S1	2.31	0.1063	2.38	0.0925
	R2	2.26	0.0272	2.33	0.0669
	S2	2.31	0.1305	2.35	0.0829
	R3	2.30	0.0567	2.33	0.0393
V-STI	R1	2.31	0.0571	2.33	0.0593
	S1	2.51	0.1025	2.41	0.0892
	R2	2.36	0.0262	2.42	0.0644
	S2	2.54	0.1258	2.51	0.0799
	R3	2.40	0.0547	2.37	0.0378

Note: M = mean; SE = standard deviation; R = resting; S = stimulation.

**Table 4 biomedicines-11-03203-t004:** Group differences (Yuen’s bootstrapped) in mean coherency between HbO concentration change (left and right hemispheres) and EMG (left and right external oblique) for the different frequency bands and stimulation phases.

	VLFO		LFO	Respiration	Cardiac
	0.01 to 0.02 Hz (scale 45–65)	0.02 to 0.04 Hz (scale 25–45)	0.04–0.15 Hz (scale 6–25)	0.16–0.6 Hz (scale 2.5–6.25)	0.8–1.5 Hz (scale 2–2.5)
Left HbO—m1R	ns	S1: Non-STI > V-STI *	ns	S2: Non-STI > V-STI **	ns
Left HbO—m2R	R2: V-STI > Non-STI *	S1: Non-STI > V-STI *	S1: Non-STI > V-STI *	R1: Non-STI > V-STI *	R3: V-STI > Non-STI *
Right HbO—m1R	ns	R2: V-STI > Non-STI *	ns	S1: Non-STI > V-STI *	ns
Right HbO—m2R	S1: Non-STI > V-STI *	ns	ns	ns	ns

* *p* < 0.05; ** *p* < 0.01; m1R = left external oblique; m2R = right external oblique; VLFO = very-low-frequency oscillations; LFO = low-frequency oscillations; ns = non significant.

## Data Availability

Data are held securely by the research team and may be available upon reasonable request and with relevant approvals in place.

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
