# Peer review of "Synergy of Muscle and Cortical Activation through Vojta Reflex Locomotion Therapy in Young Healthy Adults: A Pilot Randomized Controlled Trial"

_biomedicines, 2023, doi:10.3390/biomedicines11123203_

Round 1

Reviewer 1 Report

Comments and Suggestions for Authors

First, there is a lack of historical background explanation on the Vojta therapy.

I wonder whether Vojta therapy is significant for individuals without any medical conditions. Why was this study conducted solely on healthy people and not on patients who might benefit from the therapy?

Regarding the age of the target population, it is unclear why the study has specifically focused on this particular age group. Please provide an explanation.

The efficacy of the current intervention necessitates the inclusion of numerous additional factors to consider, as well as similarities and constraints when comparing a healthy child to one with cerebral palsy who has undergone the treatment.

Author Response

First, there is a lack of historical background explanation on the Vojta therapy.

Thank you very much for your comment. A short paragraph has been added to put the therapy in historical context.

I wonder whether Vojta therapy is significant for individuals without any medical conditions. Why was this study conducted solely on healthy people and not on patients who might benefit from the therapy?

 Thank you very much for your comment. The truth is that the question raised is really interesting. Vojta Therapy unlike many other therapies developed in health sciences has been developed based on empirical demonstration and clinical evidence. However, the neurophysiological evidence by which these clinical improvements occur is still unknown. Many studies (1,2,3) that have attempted to demonstrate neurophysiological evidence are done in healthy subjects. The main reasons why these studies should be performed previously in the population are the following:

- Ethics and safety: before applying any treatment in individuals with medical conditions, it is crucial to make sure that the approach is safe and ethical. Performing preliminary studies in healthy subjects allows the identification of potential risks and safety issues before exposing people with pathologies to any intervention.

- Determine the Baseline: Establishing a baseline in healthy subjects helps to understand the normal functioning of the organism and provides a reference point for evaluating the effects of therapy. This facilitates the identification of specific changes and improvements associated with the intervention compared to baseline.

- Variability of Individual Responses: Individuals with pathologies may have variable responses to a therapy due to the complexity of their health conditions. By working with healthy subjects, variability in responses can be reduced, making it easier to detect consistent patterns of effectiveness.

- Scientific Validation: Scientific validation of a treatment involves conducting randomized controlled studies that objectively demonstrate its efficacy. Starting with healthy subjects allows establishing the basis for designing more solid studies and better understanding the possible mechanisms of action before moving on to populations with pathologies.

- Reproducibility: Demonstrating effectiveness in healthy subjects and replicating the results in different contexts and populations contributes to the credibility and reliability of the therapy. Reproducibility is essential to ensure that results are consistent and applicable in diverse clinical situations.

In summary, conducting preliminary research in healthy subjects before applying a therapy in individuals with pathologies is an ethical and scientific practice that allows us to ensure safety, establish a solid foundation and validate the efficacy of the therapeutic approach.

Nevertheless, from our research team we plan to conduct future studies involving individuals with different neurological pathologies.

  1. Pérez-Robledo F, Sánchez-González JL. Electromyographic Response of the Abdominal Muscles and Stabilizers of the Trunk to Reflex Locomotion Therapy (RLT). A Preliminary Study. J Clin Med. 2022 Jul 3;11(13):3866. doi: 10.3390/jcm11133866. PMID: 35807151; PMCID: PMC9267217.
  2. SanzEsteban I,  Cano-de-la-Cuerda R, San-Martín-Gómez A, Jiménez-Antona C et al. Cortical activity during sensorial tactile stimulation in healthy adults through Vojta therapy. A randomized pilot controlled trial. J NeuroEng Rehabil 2021 Jan 21; 18 (1):13.
  3. Sanz-Esteban I, Calvo-Lobo C, Rios-Lago M, et al. Mapping the human brain during a specific Vojta’s tactile input: The ipsilateral putamen’s role. Medicine (United States). 2018;97(13):e0253.

Regarding the age of the target population, it is unclear why the study has specifically focused on this particular age group. Please provide an explanation.

Thank you very much for your comment. The main reason why this age range was selected was because in order to obtain healthy volunteers for the study, an appeal was made to the university community and the sample analyzed belonged to this age range. We did not want to select older subjects in order to try to have as homogeneous a sample as possible.

The efficacy of the current intervention necessitates the inclusion of numerous additional factors to consider, as well as similarities and constraints when comparing a healthy child to one with cerebral palsy who has undergone the treatment.

Thank you very much for your comment. Indeed, the effectiveness of the current intervention requires many additional factors and limitations that do not allow a correct comparison between a healthy child and a child with cerebral palsy. That is why this suggestion has been added in the limitations section.In addition, it is important to note that one of the objectives is to demonstrate the effectiveness of the intervention and the brain regions activated during stimulation. Therefore, this pilot study aims to establish the neurophysiological foundations that can be objectively observed when conducting therapy in healthy subjects and correlate them with significant signs of improvement observed as a result of clinical work with children with cerebral palsy. It is a challenge to compare healthy children with those with cerebral palsy due to numerous limitations encountered in the process.

Reviewer 2 Report

Comments and Suggestions for Authors

The study presented an investigation into the brain activation produced by Vojta Therapy, a neurorehabilitation therapy that activates reflex movement patterns. The authors recruited a sample of 27 healthy subjects and conducted an experimental study with two conditions: Vojta stimulation according to the protocol and a control non-stimuli condition. Abdominal muscle activation was measured using surface electromyography, while the activation of the cortex was assessed through near-infrared spectroscopy.

The results indicated an interaction between the stimulation phase and group in relation to the oxygenated hemoglobin concentration (HbO). Specifically, the Vojta stimulation group exhibited an increase in concentration from the baseline phase to the first resting period in the right hemisphere, contralateral to the stimulation area. This increase coincided with enhanced wavelet coherence between the HbO concentration and the electromyography (EMG) signal within the gamma frequency band during the first resting period.

The study is well designed and the paper is well written. But I still have some minor suggestions:

1 In the title, the authors should mention the study was conducted in young healthy adults but not patients.

2 I am not familiar with Vojta therapy. Does it apply manual therapy or electrical stimulation? If there was electrical stimulation, the EMG signals can be affected. If only manual therapy, it was not sure whether it produced movement artifacts to the fNIRS and EMG recording. The authors need to specify the details.

3 Why only select male subjects? This leads to a limitation of the current experiment. 

4 HbO data, there was huge baseline difference in the R1 over the right side

5 If the data have been presented in figures. there is no need to report the data again in the tables.

Author Response

The study is well designed and the paper is well written. But I still have some minor suggestions:

1 In the title, the authors should mention the study was conducted in young healthy adults but not patients.

Thank you very much for your comment. The title of the research has been modified.

2 I am not familiar with Vojta therapy. Does it apply manual therapy or electrical stimulation? If there was electrical stimulation, the EMG signals can be affected. If only manual therapy, it was not sure whether it produced movement artifacts to the fNIRS and EMG recording. The authors need to specify the details.

Thank you very much for your comment. Vojta therapy involves the application of sustained manual stimulation by exerting peripheral pressure on a specific body area. During the application of the therapy there was no movement artifact that could influence the fNIRS and EMG recording.

3 Why only select male subjects? This leads to a limitation of the current experiment. 

Thank you very much for your comment. Indeed you are right and selecting only male subjects leads to a limitation of the study that has been added in the limitations section. The present investigation is only a pilot study and not a large scale clinical trial. Near-infrared spectroscopy (fNIRS) recording requires a great deal of skill and prior learning. The persons in charge of performing the intervention and handling the recordings (JLSG, EDV and FPR) before performing the intervention had to learn how to handle and use the fNIRS recording. The fNIRS recording is very sensitive and during the learning process it was observed that people with very long hair had problems with the recording and generated a lot of noise in the signal. In order for the results reported in this pilot study to be as pure and clean as possible, the decision was made to select male subjects (due to their hair conditions). We are aware of the limitation that this implies and for future studies we will take into account and will try to expand to a larger population.

4 HbO data, there was huge baseline difference in the R1 over the right side

We appreciate the reviewer's insightful observation regarding the aparent difference in the baseline measure of HBO in our study. In any case, as reported in the paper, robust paired t-tests (Yuen's bootstrapped) showed no significant differences between groups at R1-right hemisphere [t(8.9)=1.1; p = .34]. In any case, this observation about the HBO baseline measure is particularly relevant in the context of our study's main limitation, which is the reliance on a single measurement session using near-infrared spectroscopy. This is acknowledged in the article. Due to constraints in our study design and the practicalities of using NIRS, we were limited to a single measurement session/trial for each subject. This limitation is important to consider when interpreting the variability observed in the HBO measures and, also, potentially impacts the generalizability of our findings, as multiple measurement trials or sessions might have provided a more comprehensive understanding of HBO variability. Several factors might contribute to the observed discrepancy in HBO baseline measures, including biological variability among subjects and potential technical nuances specific to HBO measurement and the fact that mean hbo concentrations were calculated in the full time periods. We recognize the importance of continually refining research methodologies and appreciate the reviewer’s feedback, which underscores areas for improvement in our study and future research endeavors.

5 If the data have been presented in figures. there is no need to report the data again in the tables.

Thank you for your feedback regarding the presentation of data in both figures and tables. We understand the concern about potential redundancy. Our intent in providing the data in both formats was to cater to different reader preferences and to enhance the clarity and accessibility of the information. Figures offer a visual representation that can be immediately impactful and intuitive, especially for understanding trends and relationships in the data. On the other hand, tables provide precise numerical values, which are essential for readers interested in specific details or for those who may wish to perform further calculations. Having said this, alternatively, we can include figures in the main manuscript and relegate tables to supplementary material. This approach ensures that all data are available without overwhelming the main text. We will be ready to  follow the instructions provided by the reviewer or the editors in this regard.

Round 2

Reviewer 1 Report

Comments and Suggestions for Authors

The authors revised the manuscript extensively.